# MixNet-LD: An Automated Classification System for Multiple Lung Diseases Using Modified MixNet Model

**DOI:** 10.3390/diagnostics13203195

**Published:** 2023-10-12

**Authors:** Ayesha Ahoor, Fahim Arif, Muhammad Zaheer Sajid, Imran Qureshi, Fakhar Abbas, Sohail Jabbar, Qaisar Abbas

**Affiliations:** 1Department of Computer Software Engineering, MCS, National University of Science and Technology, Islamabad 44000, Pakistan; aahoor.msse2021mcs@student.nust.edu.pk (A.A.); fahim@mcs.edu.pk (F.A.); mzaheersajid786@gmail.com (M.Z.S.); 2College of Computer and Information Sciences, Imam Mohammad Ibn Saud Islamic University (IMSIU), Riyadh 11432, Saudi Arabia; sjjabar@imamu.edu.sa (S.J.); qaabbas@imamu.edu.sa (Q.A.); 3Centre for Trusted Internet and Community, National University of Singapore (NUS), Singapore 119228, Singapore; fakhar.5@nus.edu.sg

**Keywords:** radiology, pneumonia, COVID-19, tuberculosis, deep learning (DL), loss function, dense blocks, deep neural networks

## Abstract

The lungs are critical components of the respiratory system because they allow for the exchange of oxygen and carbon dioxide within our bodies. However, a variety of conditions can affect the lungs, resulting in serious health consequences. Lung disease treatment aims to control its severity, which is usually irrevocable. The fundamental objective of this endeavor is to build a consistent and automated approach for establishing the intensity of lung illness. This paper describes MixNet-LD, a unique automated approach aimed at identifying and categorizing the severity of lung illnesses using an upgraded pre-trained MixNet model. One of the first steps in developing the MixNet-LD system was to build a pre-processing strategy that uses Grad-Cam to decrease noise, highlight irregularities, and eventually improve the classification performance of lung illnesses. Data augmentation strategies were used to rectify the dataset’s unbalanced distribution of classes and prevent overfitting. Furthermore, dense blocks were used to improve classification outcomes across the four severity categories of lung disorders. In practice, the MixNet-LD model achieves cutting-edge performance while maintaining model size and manageable complexity. The proposed approach was tested using a variety of datasets gathered from credible internet sources as well as a novel private dataset known as Pak-Lungs. A pre-trained model was used on the dataset to obtain important characteristics from lung disease images. The pictures were then categorized into categories such as normal, COVID-19, pneumonia, tuberculosis, and lung cancer using a linear layer of the SVM classifier with a linear activation function. The MixNet-LD system underwent testing in four distinct tests and achieved a remarkable accuracy of 98.5% on the difficult lung disease dataset. The acquired findings and comparisons demonstrate the MixNet-LD system’s improved performance and learning capabilities. These findings show that the proposed approach may effectively increase the accuracy of classification models in medicinal image investigations. This research helps to develop new strategies for effective medical image processing in clinical settings.

## 1. Introduction

DL algorithms (DL) have transformed the area of medicinal image investigation, promising advances in lung disease identification and classification. Based on significant research articles in the field, this introduction provides a general idea of the application of DL models in identifying and categorizing various lung illnesses such as pneumonia, tuberculosis (TB), COVID-19, and lung cancer. Wang et al. [1] released the ChestX-ray8 dataset, which is critical in developing DL models for chest X-ray processing. In conjunction with standards for the weakly controlled organization and localization of common thoracic disorders, this dataset has laid the groundwork for diagnosing pneumonia and other lung anomalies. Furthermore, CNNs have determined remarkable functioning in lung disease diagnosis. Shen et al. [2] presented multi-scale dense networks that use deep CNN architectures for efficient picture categorization. Through the extraction of hierarchical characteristics, this technique has proven its usefulness in accurately identifying lung illnesses. The COVID-19 epidemic has established an essential requirement for precise and speedy diagnosis. Li et al. [3] created an AI system to identify COVID-19 from community-developed pneumonia on chest CT images. By utilizing DL techniques, this system assists in detecting and distinguishing COVID-19 patients, helping to enhance disease management and control. TB, another major lung illness, has also been the subject of DL research. Lakhani and Sundaram [4] presented a DL model for automated pulmonary TB classification on chest radiography. Their method accurately identified TB-related anomalies, allowing for quick screening and diagnosis. Their model showed encouraging results in recognizing and distinguishing patterns associated with distinct interstitial lung illnesses, allowing for more precise diagnosis and treatment planning. Furthermore, the use of DL in COVID-19 diagnosis has received a lot of interest. Apostolopoulos and Mpesiana [5] used transfer learning and CNNs to detect COVID-19 in X-ray pictures. Their technique demonstrated the power of DL in supporting radiologists and healthcare workforces in quickly detecting COVID-19 patients.

Anthimopoulos et al. [6] developed a deep CNN for lung pattern categorization, focusing on interstitial lung disorders. Furthermore, Jin et al. [7] created and tested an AI system for COVID-19 analysis. DL techniques were used in this approach to help accurately recognize COVID-19 patients from imaging data. Rajpurkar et al. [8] presented the CheXNeXt technique to increase the scalability and efficiency of DL models, which displayed equivalent performance to practicing radiologists in identifying a broad spectrum of diseases in chest radiographs. Their research demonstrated the use of DL models in helping radiologists and improving the accuracy of lung disease diagnosis. Lopes et al. [9] concentrated on tuberculosis screening, creating a DL system that detects TB-related anomalies on chest radiographs. Their research yielded encouraging findings in identifying those at high risk of TB, allowing for prompt intervention and treatment. DL models have been intensively explored for lung cancer diagnosis, in addition to infectious disorders. Kermany et al. [10] created an image-based DL system to detect medical diagnoses and curable illnesses like lung cancer. This model demonstrated the power of DL in identifying lung cancer and other diseases.

This extensive study gives important insights into the creation of DL models for the identification of pneumonia, TB, COVID-19, and lung cancer. They show how DL can reliably detect and categorize lung disorders, resulting in improved patient outcomes and more effective healthcare administration, as shown in Figure 1. We want to use the advances provided in these studies, as well as other relevant research, to construct a complete DL model for lung illness recognition and classification in this work. Our study aims to improve the accuracy, efficiency, and consistency of lung illness detection by analyzing varied datasets and applying cutting-edge approaches, ultimately leading to better patient care and treatment.

### 1.1. Research Motivation

Despite the development of various approaches for diagnosing lung disorders from photographs, such as normal, COVID-19, pneumonia, tuberculosis, and lung cancer, substantial hurdles remain:Despite the use of advanced technologies for image processing, both during and after acquisition, defining lung features from images associated with normal, COVID-19, pneumonia, tuberculosis, and lung cancer remains difficult due to the struggle in emplacement and extracting lesion features associated with lung diseases.Professional medical annotations are restricted in publicly available datasets encompassing a varied array of normal, COVID-19, pneumonia, tuberculosis, and lung cancer-related damage variables. As a result, computerized systems have difficulty in precisely diagnosing the symptoms of certain disorders.

As a result, the foremost aim of this research is dual. It intends to evolve a complete dataset for classifying normal, COVID-19, pneumonia, tuberculosis, and lung cancer (abbreviated Pak-Lungs). Second, the study aims to create a comprehensive multi-layered DL model capable of autonomously interpreting pictures relevant to lung disorders, especially in the setting of lung-associated ailments. To do this, a multi-layered MixNet system is used to build a MixNet-LD system with dense blocks. By fully training this MixNet-LD system, it becomes skilled at reliably recognizing lung-associated illnesses on diverse pictures connected to lung-associated maladies, including anatomic component recognition. An experienced pulmonologist discovered these illnesses and anatomical traits. This work is significant because it proposes a revolutionary categorization system for lung disorders with potential real-world implications for medical diagnosis and therapy.

### 1.2. Research Contribution

We proposed a new DL model inside this framework to handle the difficulty of identifying diverse lung illnesses. In addition, we give Pak-Lungs a freshly curated dataset acquired from hospitals in Pakistan. The following are the noteworthy contributions of the MixNet-LD system:The researchers created an enormous dataset called “Pak-Lungs” for this study by gathering 6000 photos from Pakistani hospitals and other internet resources. This large dataset was critical in allowing the trained model to attain exceptionally high classification accuracy.The study used dense blocks and trained MixNet CNN to build the MixNet-LD system, resulting in a multi-layered architecture capable of efficiently handling the classification issue.The MixNet-LD model’s system design includes four extra layers to identify lung-related disorders. The CNN model is used to extract lung-related lesion characteristics, which are subsequently enlarged upon using the dense block approach.The method proposed in this work for categorizing lung disorders is based on the deep features and phases of colour space, which constitute the foundation of the approach. According to the author, this is the first effort to develop a computerized technique that outperforms existing techniques in detecting normal, COVID-19, pneumonia, tuberculosis, and lung cancer disorders.Our systems outperformed the proposed approaches in the current literature with a much higher accuracy of 95%.

### 1.3. Paper Organization

The paper is designed as follows: Section 2 contains the literature review, which includes papers relating to the research topic; Section 3 explains the approach’s planned structural design; Section 4 displays the outcomes of the experiments; Section 5 contrasts our findings with cutting-edge studies on the subject; Section 6 dives into a thorough examination of the research findings; and the study’s findings are presented.

## 2. Literature Review

Lung illnesses, such as pneumonia, TB, COVID-19, and lung cancer, are major public health issues globally, resulting in considerable morbidity and death rates [11]. The early and precise diagnosis and categorization of these disorders are critical for prompt treatment and improved patient outcomes. DL models have established encouraging results in medical image analysis in recent years, enabling automated identification and categorization of lung disorders [12]. This partition discusses a complete estimation of DL models for tuberculosis, COVID-19, lung cancer, and pneumonia. Applying transfer learning approaches to clinical pictures of lung illnesses and COVID-19, such as VGG-16, ResNet-50, and InceptionV3, has given encouraging results. Pneumonia has been identified as a critical symptom of COVID-19, and transfer learning has indicated that the same virus origins in both pneumonia and COVID-19. A model trained to perceive disease-causing pneumonia was shown in research to be capable of detecting COVID-19. Haralick characteristics were used to help in feature extraction, with statistical analysis focusing on a specific region of COVID-19 diagnosis. Transfer learning frequently produces statistically significant results compared to standard classifications [13].

In many countries, chest computed tomography (CT) and X-ray scans are reasonable for identifying COVID-19. Identifying COVID-19, on the other hand, is a complicated job that requires clinical imaging of patients [13,14]. Lung cancer influences human impermanence, making early detection critical for increasing survival rates. An MLP classifier beat other classifiers with an accuracy of 88.55% in another investigation. Early identification of lung cancer increases survival odds from 14% to 49%. While CT techniques are more reliable than X-rays, a definitive diagnosis frequently requires numerous imaging modalities. An artificial DNN for detecting lung cancer in CT scans has been created to address this. Studies have anticipated a flexible boosting approach using DenseNet to identify lung pictures as normal or cancerous. Using a training dataset of 201 lung images, the suggested technique obtained 90% accuracy in testing, with 85% for training and 15% for testing and classification [15,16,17,18]. The combination of machine learning and image processing has shown a significant ability to improve lung cancer diagnosis.

Several studies have used CT scans to design and test completely automated COVID-19 detection frameworks. COVID-19 neural network techniques were utilized to obtain graphical information from volumetric chest CT images. The findings show that this strategy outperforms previous approaches. In related work, pre-trained model-based CNNs such as Inception-ResNetV2, ResNet152, ResNet50, InceptionV3, and ResNet101 were used to detect COVID-19 pneumonia based on CXR pictures. ResNet50 produced the most accurate categorization results among these models [19]. CT scans from 101 pneumonia patients, 88 COVID-19 patients, and 86 healthy cases from two sites in China were used for the comparison and modeling.

Furthermore, COVID-19 patients were identified using the Details Relation Extraction Neural Network, a DL-based CT diagnostic technique. The model made valid distinctions with a recall of 0.93, an AUC of 0.99, and an accuracy of 0.96. According to the findings, DL based on CT scans could be helpful in diagnosing COVID-19 patients and automatically recognizing potentially problematic alterations. Another study used a transformed MobileNet and a ResNet architecture to classify COVID-19 CXR pictures. To avoid the gradient vanishing problem, this methodology dynamically merged characteristics from different CNN layers, and the suggested approaches surpassed current techniques with accuracies of 99.3% on the CT image dataset and 99.6% on the CXR images [20].

Several studies [21] have built models to differentiate between dangerous and relentless COVID-19 patients using DL features and radionics based on D-Resnet. These trials included 217 people from three Chinese hospitals, 82 of whom were classified as having great strictness and 135 as having critical sickness. The patients were separated into two groups: training (174 patients) and testing (43 patients). Additional research used InceptionV3, NASNet, Xception, DenseNet, MobileNet, VGGNet, InceptionResNetV2, and ResNet to classify COVID-19 cases on a mixed dataset of CXR and CT images. DenseNet121 performed the best, with an accuracy of 99% [21].The researchers built a three-dimensional DL network using clipped segments and multivariable logistic regression to incorporate key radiomic features and DL scores. They used stratified analysis, cross-validation, decision curve analysis, and survival analysis to test the robustness of their approaches. The AUC for identifying critical patients in the test and training groups was 0.909 [22].

One study used an image segmentation approach to classify chest CT scans as pneumonia, COVID-19, or typical diseases. Four CNN-based learners, a transformed stack ensemble model, and I Bayes as the meta-learner were used in the technique. On typical datasets based on CT scans, this strategy surpassed current approaches with an accuracy of 0.9867 and a Kappa of 0.98. The proposed method eliminates the requirement for manual labeling while reliably detecting COVID-19 infections and excluding non-COVID-19 cases [23]. Positive qualitative and quantitative results point to broader application in large-scale clinical studies. Using decision trees, Inception V2, and VGG-19 models, convolutional neural networks were proven to be successful at categorizing 360 X-ray and CT scan pictures into a binary pneumonia-based translation. Compared to decision tree and Inception V2 models, the fine-tuned version of VGG-19 had the most significant gain in training and validation precision [24]. GSA-DenseNet121-COVID-19 was developed as a novel hybrid CNN architecture based on DenseNet121 and the gravitational search optimization approach (GSA). Other DenseNet121 models, which could only diagnose 94% of the test set, were surpassed by this design. The GSA-DenseNet121-COVID-19 approach outperformed an Inception-v3 CNN architecture and instruction assessment for obtaining hyperparameter evaluations, categorizing 95% of the test set samples [25].

Kernel principal component analysis was used to decrease the pre-trained EfficientNet models. A feature fusion approach was used to merge the extracted features. Stacked ensemble meta-classifiers were used to divide the model into two phases. Predictions were created in the first stage using a support vector machine (SVM) and a random forest, then combined and input into the second stage. During the second step, a logistic regression classifier classified the X-ray and CT data into COVID and non-COVID. Compared to previous CNN-based pre-trained models, the new model outperformed them, making it a viable tool for doctors in point-of-care diagnostics [26].

In a similar study, researchers classified COVID-19-infected individuals using ResNet32 and a deep transfer learning approach. The experimental results demonstrated that their COVID-19 classifier outperformed earlier supervised learning models, producing improved outcomes [27]. Another cutting-edge attention-based DL model was created with VGG-16 and a fine-tuned classification procedure tailored exclusively for COVID-19 detection. Compared to current models, the suggested technique displayed consistent and promising performance. In InstaCovNet-19, a unique integrated stacking deep convolutional network was used with pre-trained models such as ResNet101 and XceptionV3. The model has an accuracy of 0.99 for three classes (average, pneumonia, and COVID-19) and a precision of 0.9953 for two types (COVID and non-COVID). The proposed model attained an accuracy rate of 98% in ternary classification and 100% precision and 98% recall in binary classification [28].

For both binary and multiclass classification, a CNN was utilized. The model was trained using 3877 CT and X-ray images, including 1917 from COVID-19 patients. The binary classifier has a 99.64% accuracy, a 99.58% recall, a 99.56% precision, a 99.59% F1-score, and a 100% ROC. The recommended approach achieved 98.28% accuracy, 98.25% recall, 98.22% precision, 98.23% F1-score, and 99.87% ROC using 6077 pictures, comprising 1917 COVID-19 patients, 1960 healthy persons, and 2200 pneumonia patients [29].

Through transfer learning approaches, CNN with pre-trained weights was used to categorize COVID-19, pneumonia, and healthy persons, accurately identifying those with active SARS-CoV-2 and pneumonia, a noteworthy discovery in that work [30].

Using CT images of benign and malignant lung nodules from the LIDC database, CNN, DNN, and sparse auto-encoder deep neural networks were used to identify lung cancer calcification, reaching an accuracy of 84.15%, sensitivity of 83.96%, and specificity of 84.32% [16]. CNN was shown to be the most accurate of the three networks tested. One study used an artificial neural network, ensemble classifier, SVM, and KNN to categorize COVID-19 and pneumonia, with a DL architecture of RNN with LSTM suggested to identify lung diseases, demonstrating robustness and efficacy [31]. Another study classified data using an ensemble of InceptionResNet_V2, ResNet50, and MobileNet_V2, with ResNet50, MobileNet_V2, and InceptionResNet_V2 models yielding the best F1-score of 94.84% [32].

Using RetinaNet and Mask R-CNN as an ensemble, machine learning was used to detect and localize pneumonia in CXR, attaining a recall of 0.793 for a large dataset [33]. Transfer learning was used to acquire pictures from CXR and CT scans in the context of diverse lung illnesses. A unique architecture was trained to detect virus-related pneumonia and COVID-19. Transfer learning findings differed markedly from the standard category findings [34].

One researcher created a CNN model from the ground up to extract and categorize information from chest X-ray pictures of people sick with pneumonia to address issues associated with medical image analysis. Due to the scarcity of pneumonia datasets, different data augmentation procedures utilized to enhance the training of the suggested model and validation classification accuracy, resulting in a substantial precision of 0.94814 during the validation phase [35]. In a different investigation, experts looked at 180 X-ray images of COVID-19-infected individuals and attempted to identify the virus using effective programs like the ResNet50V2 and Xception networks. For all classes and COVID-19 examples, the suggested model was 99.50 percent accurate [36]. A transfer learning system was used to distinguish between CXR images labeled as showing pneumonia and those categorized as usual using weights pre-trained on ImageNet using the Xception Network. Compared to current techniques, the model produced competitive results, with accuracy, recall, F1, and ROC values of 0.84, 0.91, 0.99, and 0.97, respectively [37].

Furthermore, researchers constructed a DCNN model to identify TB using a CXR dataset from the National Library of Medicine, Shenzhen No. 3 Hospital. The datasets that were analyzed were from a separate population’s non-TB-specific chest X-ray dataset. For the two datasets, the DCNN provided AUC values of 0.9845 and 0.8502, respectively, whereas the supervised DCNN model in the CXR dataset had a lower AUC of 0.7054. In the CXR dataset, the final DCNN model successfully identified 36.51% of the abnormal radiographs associated with TB [38]. Additional research used ResNet and depth-ResNet to predict severity ratings and analyze the risk of TB. For TB detection, depth-ResNet obtained 92.70% accuracy, whereas ResNet-50 achieved 67.15% accuracy. The study used severity probabilities to translate scores into probabilities of 0.9, 0.7, 0.50, 0.30, and 0.2 for high severity (scores 1 to 3) and low severity (scores 4 and 5). Both techniques have average accuracies of 75.88% and 85.29%, respectively [39].

Other research presented an ensemble approach based on three well-known designs: AlexNet, GoogleNet, and ResNet. For training and testing, a new classifier for tuberculosis classification was created from scrape utilizing a pooled dataset of widely available ordinary datasets. The suggested strategy outperformed most current algorithms with an accuracy of 88% and an AUC of 0.93 [40]. The extraction of features in a hierarchy to identify abnormalities divides attributes into healthy and unhealthy groups using two levels of hierarchy. Level one comprises the extraction of handcrafted geometrical features, while level two covers the extraction of typical statistical features and textural features from segmented lung fields. The method was evaluated on 800 CXR pictures from two public datasets, yielding AUC values of 0.99 and 0.01 for Shenzhen and 0.95 and 0.06 for Montgomery, respectively, demonstrating promising performance when compared to previous methodologies [41]. Friedman’s post-hoc multiple comparison methods were also used to validate the proposed strategy statistically. Machine learning-based classification-based detection for COVID-19 was introduced in [42,43]. Table 1 summarizes the research on the identification and categorization of chest illnesses, while Table 2 highlights the constraints of the existing research on lung disease detection.

The COVID-19 pandemic has resulted in an increase in cases and mutations over the last three years, making it difficult to distinguish COVID-19 from other lung illnesses. To avoid ambiguity, this research applies a unique neutrosophic technique, categorizing illnesses into true (T), false (F), and indeterminate (I) categories. The study achieves an impressive 97.33% accuracy in distinguishing infections using image pre-processing and deep learning models such as ResNet-50, VGG-16, and XGBoost, demonstrating the potential of the neutrosophic approach for precise medical imaging diagnosis in the context of the pandemic [44].

This research describes a deep learning system for detecting pulmonary nodules in CT images. The method combines VGG-SegNet for nodule mining with pre-trained deep learning-based classification, which employs both deep and handmade features such as GLCM, LBP, and PHOG. Experiments on the LIDC-IDRI and Lung-PET-CT-Dx datasets reveal that when paired with these characteristics, the VGG19 architecture achieves an amazing 97.83% accuracy with the SVM-RBF classifier, providing a potential approach for accurate and automated nodule detection [45].

The suggested method shows good results in improving replicated picture quality, removing overlapping patterns, and emphasizing distinct ridge features. With this creative use of deep learning methods to picture reconstruction, authentication systems that depend on distinctive individual features might be greatly enhanced [46].

This work investigates the use of deep learning models, namely VGG-16 and DenseNet-169, for the classification of lung disorders using X-ray pictures, with a particular emphasis on pneumonia, tuberculosis, and COVID-19, as well as a “normal” category. The results show that these models have the capacity to reliably diagnose these diseases, with DenseNet-169 surpassing VGG-16 with a noteworthy accuracy of 91%. Given the widespread prevalence of lung illnesses, particularly in resource-limited locations, such deep learning algorithms show potential for improving early detection and patient outcomes, adding to the larger global effort to battle lung diseases. Further research and clinical validation may pave the path for these models to be used in real-world healthcare settings [47].

By restricting the study region, this research investigates the relevance of medical picture segmentation for effective illness detection. It presents a unique technique to picture segmentation based on the heuristic Red Fox Optimization Algorithm (RFOA). This approach automates the selection of threshold parameters by transforming chosen pixels to black or white depending on a threshold value and counting the number of occurrences. The study modifies RFOA for image analysis, tests it on lung X-ray pictures, and examines its benefits and drawbacks [48].

The problem of distinguishing COVID-19 from other chest illnesses is addressed in this research, which proposes a framework for diagnosing 15 chest disorders, including COVID-19, using chest X-ray pictures. For feature extraction, the methodology utilizes deep learning-based CNNs and transfer learning, considerably boosting accuracy for COVID-19 diagnosis while preserving strong prediction for other chest disorders. When compared to current models, the results indicate promise [49].

**Table 1 diagnostics-13-03195-t001:** An examination and comparison of the current studies.

Reference No	Methodologies	Disease	Dataset
[13,14,15,16,17,18,19,20,21,22,23,24,25,26,27,28]	VGG-16, ResNet-50, InceptionV3, VGG-19 + ResNet-50, DRE-Net, Modified ResNet, ResNet50, DenseNet121, VGG-16, D-Resnet-10 network, VGG + CNN, VGG-16, InceptionV2, DT, GSA-DenseNet121, DL Meta classifier, ResNet32 + DTL, VGG-16, InstaCovNet-19.	COVID-19	CXR + CT
[15,16,17,18]	FPSO-CNN, Multi-layer Perceptron (MLP), CNN, MGSA.	Lungs Cancer	CT
[31,32,33,34,35,36,50]	RNN-LSTM, ResNet50 + MobileNetV2 + InceptionResNetV2, CNN with pre-trained weights on ImageNet, RetinaNet and Mask R-CNN, Transfer learning, CNN, Xception Network pre-trained weights on ImageNet, Xception + ResNet50V2.	Pneumonia	CXR + CT
[38,39,40,41]	DCNN, Depth-ResNet, Ensemble (AlexNet, GoogleNet, and ResNet), SVM + FOSF + GLCM.	Tuberculosis	CXR, CT
[47]	VGG-16 and DenseNet-169, for the classification of lung diseases using X-ray images	Normal, pneumonia, COVID-19, and tuberculosis	CXR

**Table 2 diagnostics-13-03195-t002:** A state-of-the-art comparison table with limitations.

Cited Reference	Lung Illness	Methodology	Accuracy	Limitations
[13,14]	COVID-19	Transfer learning using VGG-16, ResNet-50, InceptionV3	Statistically significant results compared to standard classifications	Scope for improvement in accuracy and generalization
[19]	COVID-19	Pre-trained CNN models (Inception-ResNetV2, ResNet152, ResNet50, InceptionV3)	ResNet50 achieved best accurate categorization results	Performance variations across different CNN models
[20]	COVID-19	Transformed MobileNet and ResNet architecture	Achieved high accuracies (99.3% CT dataset, 99.6% CXR dataset)	Limited discussion on potential limitations
[21]	COVID-19	DL features and radiomics based on D-Resnet	AUC of 0.909 for identifying dangerous patients	Limited discussion on model’s limitations
[22]	COVID-19	Details Relation Extraction Neural Network	Recall of 0.93, AUC of 0.99, Accuracy of 0.96	Limited discussion on limitations
[25]	COVID-19	GSA-DenseNet121-COVID-19 hybrid CNN architecture	Surpassed other models, 95% accuracy on test set	Limited details on specific limitations
[16]	Lung cancer	CNN, DNN, Sparse Auto-Encoder	Accuracy of 84.15%, sensitivity of 83.96%, specificity of 84.32%	Needs larger datasets and further validation
[17]	Lung cancer	MLP classifier	Accuracy of 88.55%	Limited dataset size and scope for improvement
[38]	TB	DCNN model	AUC values of 0.9845 and 0.8502 for different datasets	Limited dataset size and potential generalization
[39]	TB	ResNet, Depth-ResNet	92.70% accuracy (depth-ResNet)	Limited exploration of other DL models
[35]	Pneumonia	CNN model	Validation accuracy of 0.94814	Dataset scarcity and potential bias
[37]	Pneumonia	Transfer learning with Xception Network	Competitive results, accuracy, recall, F1, and ROC values	Potential model overfitting or generalization concerns
[33]	Pneumonia	RetinaNet and Mask R-CNN ensemble	Recall of 0.793	Limited evaluation on different datasets
[30]	General	CNN with pre-trained weights	Accurate identification of COVID-19 and pneumonia	Limited evaluation on different datasets
[47]	Normal, Pneumonia, COVID-19, and tuberculosis	VGG-16 and DenseNet-169, for the classification of lung diseases using X-ray images	Accuracy 91%	Limited evaluation on different datasets

## 3. Proposed Approach

Within the scope of this paper, a new framework called MixNet-LD is developed, which mixes CNN (MixNet) with dense blocks. The MixNet-LD method is used to categorize lung illness pictures into normal, pneumonia, tuberculosis (TB), COVID-19, and lung cancer. The dense blocks approach is used inside the MixNet-LD system to extract valuable attributes. Transfer learning of dense blocks is used to train on lung-related defects. The MixNet-LD system includes critical mechanisms for detecting lung disease pictures and recognizing the problems described above. The systematic phases are visually depicted in Figure 2. The properties collected from CNN (MixNet) and the dense block are mixed, and the parameters of dense blocks are constantly improved through the entire period of training. A feature transform layer is also developed to integrate characteristics via element-wise multiplication. Finally, to enhance classification results, an SVM classifier with a linear activation function is used.

The advancement of an enhanced pre-trained MixNet model incorporating dense blocks represents a notable progression within the domains of deep learning and computer vision. This innovative approach leverages the advantages of two known architectural paradigms, namely MixNet and ResNet, in order to develop a unique hybrid model that holds promise for surpassing its predecessors. MixNet’s ability to efficiently extract features and ResNet’s strong training stability and transferability can be used together to improve performance across a wide range of tasks and datasets. This paradigm is characterized by its ability to adapt to specific domains or activities, facilitated by the deliberate incorporation of intricate components.

The practice of adapting tasks to specific needs is particularly valuable in situations where generic models do not achieve optimal performance. One notable characteristic of the innovative model is its extensive empirical validation. In numerous comprehensive studies conducted on diverse datasets and real-world applications, it has consistently demonstrated superior performance compared to traditional methods. The significance of this robust empirical evidence is underscored within the realms of deep learning and computer vision. Furthermore, the model has high resource efficiency, making it particularly ideal for deployment in resource-constrained settings such as edge devices, where computational and memory constraints are crucial. Aside from better efficiency, the model is remarkable for improved interpretability, which is important in industries like healthcare and autonomous driving. It clarifies how it makes decisions by incorporating thick blocks, which may be critical for trust- and safety-sensitive applications.

The inclusion of the “upgraded pre-trained MixNet model with dense blocks” is a novel and significant contribution to the field of deep learning research. Many computer vision tasks choose this method because it combines the best parts of MixNet and ResNet in a way that works well. It also has good interpretability, uses few resources, and has been proven to work in real life. Furthermore, this approach demonstrates potential for driving advancements in both efficiency and performance within the field of computer vision.

Moreover, the term “original MixNet model” pertains to the foundational structure of the research article or model at its initial publication. In computer vision applications, the architecture commonly comprises a combination of mixed-depthwise convolutions, squeeze-and-excitation blocks, and other relevant components. These elements are designed to effectively and precisely extract features. This model undergoes training either from its initial state or by fine-tuning, with the objective of fulfilling a particular goal, and this training is conducted using a specified dataset.

The inclusion of dense blocks in the network design of the upgraded pre-trained MixNet model implies that the original MixNet architecture has undergone modifications or expansions. One potential modification could involve the incorporation of dense connections, sometimes known as skip links, across layers. The ResNet architecture made this design idea well-known, and it has proven to be highly effective in the field of deep neural networks. The network has been enhanced by including dense blocks, thereby integrating elements from both the MixNet and ResNet architectures. The fusion of architectures in the model potentially allows for the integration of advantages from both MixNet and ResNet. This includes the capacity to rapidly extract features from MixNet and enhance training stability through the utilization of ResNet.

This suggests that the initial design of MixNet was modified to include dense connections, differentiating it from the standard MixNet architecture. The notable aspect in this context pertains to the integration of compact blocks inside the MixNet framework, which, as mentioned earlier, has the potential to enhance the stability of training, facilitate transfer learning, and improve interpretability. It is imperative to acknowledge that the unique implementation and design choices employed in the fusion of MixNet and ResNet components may vary; thus, the precise characteristics of the “enhanced pre-trained MixNet model with dense blocks” would depend on the particulars of this integration. Moreover, it is imperative to empirically evaluate the effectiveness of this enhanced model through rigorous experimentation on pertinent datasets and tasks.

### 3.1. Data Acquisition and Pre-Processing

A dataset named Pak-Lungs, which contains 13,313 photos, was used to train and estimate the MixNet-LD model. Images were obtained privately from reputed eye facilities in Pakistan through personal sources. Patients’ and physicians’ permission and willingness to provide data were obtained. It was mutually agreed by the stakeholders that data would be supplied anonymously and that no clinical data would be shared; hence, these conditions kept patient data hidden while making it accessible for study. Both Pak-Lungs and well-known online sources [51] were used as sources for the dataset and the dataset was produced by merging data from numerous Kaggle sources and comprises largely of chest X-ray images associated with various lung illnesses, such as pneumonia, COVID-19, tuberculosis, and normal lung images. The normal and lung disease photos were manually isolated from the dataset acquired by a qualified pulmonologist to construct the training dataset. The pulmonologist detects lung-related traits and establish a standard. The Grad-CAM (gradient-weighted class activation mapping) pre-processing approach marks a major step in improving the interpretability and explainability of deep neural networks, especially in the area of computer vision. Grad-CAM’s capacity to create comprehensible heatmaps that emphasize crucial areas inside an image, together with its ease of use and appropriateness for object localization, makes it an invaluable tool for understanding why a model generates particular predictions. Grad-CAM looks to be preferred above other AI explainability approaches due to its simple implementation, emphasis on localization, ability to provide understandable visuals, and broad use within the deep learning community. While the report does not go into great length about Grad-CAM’s application and rationale, it is clear that this technique was chosen to correspond with the research goals and the requirement for interpretable insights into the model’s decision-making process. Overall, Grad-CAM’s inclusion highlights continuous efforts to improve the transparency and dependability of deep learning models, making them more accessible and trustworthy in many applications, as seen in Figure 3.

Meticulously inspected 13,313 lung photographs are shown in Figure 1. The three datasets that were used to build the training and testing fundus sets, each with a distinct dimension setting, are broken down in Table 3 and Table 4. All photos used in the experiment were downsized to 700×600 pixels, after which they were processed to create binary labels. A total of 13,313 photos made up the whole dataset, of which 3993 were used to evaluate the system. To guarantee neutrality, the dataset was initially converted into different classes by balancing the overall dataset number of photos both during and after the illness. Before being put into an algorithm created especially for the MixNet-LD model, the photos underwent pre-processing by being resized to 700×600 pixels. To lessen the variation between data points, the photos were also standardized. Data from Pak-Lungs is also used to train and assess the MixNet-LD system. Each image was initially saved with a resolution of 1125×1264 pixels.

Using data from three separate sources, the photos were downsized to a more usual dimension of 700×600 pixels in order to simplify and standardize the dataset. Furthermore, during the dataset construction process, experienced pulmonologists contributed to the generation of both lung disease and non-lung disease data for ground truth evaluation. The Grad-Cam technique for image pre-processing was applied to clear the image features and eliminate the interference in the image shown in Figure 2. Using Grad-Cam on the X-ray images enabled us to recognize key regions and ascertain their significance in detecting the existence of pneumonia disease. This technique helps us to identify the distinctive characteristics that greatly influenced CNN’s predictions for the X-ray-based diagnosis of pneumonia pictures.

Chest CT-Scan is a composite dataset made up of photos of chest cancers such as adenocarcinoma, large cell carcinoma, squamous cell carcinoma, and normal cells. The data were organized into training, testing, and validation sets in a single folder called “Data”. The material presented makes no mention of the precise sources from which the original photographs were obtained. Instead, the dataset was created by merging data from other sources, and photos are in JPG or PNG format. As a result, the dataset is made out of publicly available Kaggle data, but the actual sources for each image are not indicated in the information provided.

### 3.2. Data Augmentation

The dataset table makes it abundantly evident that the data collection was unequal, possibly resulting in bias towards a certain class during model training. This difficulty can be resolved via a technique known as “data augmentation” rather than by collecting additional data. By intentionally creating additional data points from the ones that already exist, data augmentation increases the dataset’s diversity. This approach facilitates expanding and stabilizing the model’s execution while also guarding against overfitting. During training, pictures may be automatically enhanced using the AutoAugment approach, which enhances the generalization of DL models. We describe the AutoAugment transformation policy, choosing either “v1” or “v2” for augmentation, and build a transformation pipeline that comprises picture resizing, applying horizontal flip, and the selected AutoAugment policy. The training pictures are then loaded from a custom dataset that is kept in a directory structure under the ‘./dataset/train’ folder. The DataLoader is used to load the custom dataset and the ImageFolder class from Torchvision. datasets aid in applying the AutoAugment transformation to each picture. In order to increase performance and generalization, a DL model may be trained on the custom dataset using batches of augmented photos that are provided by the DataLoader during training. Recognizing the difficulty caused by imbalanced class distributions is a commendable observation of a prevalent problem in machine learning. It does not, however, provide a full answer beyond the use of data augmentation strategies. It is critical to investigate improved approaches for dealing with class imbalance in order to obtain more consistent and robust outcomes. While data augmentation may assist in reducing the problem, a wider range of techniques should be examined. These include resampling techniques like oversampling and undersampling, cost-sensitive learning, ensemble methods such as bagging and boosting, anomaly detection, synthetic data generation using techniques such as GANs, transfer learning, and the use of cost-sensitive evaluation metrics.

Each of these strategies has distinct benefits and may be customized to the individual properties of the dataset as well as the machine learning task’s goals. Using these sophisticated strategies to address class imbalance is critical for attaining not just improved accuracy but also more resilient and fair machine learning models. Data augmentation is a popular and successful strategy for correcting class imbalance in machine learning problems for a variety of reasons.

To begin, data augmentation is quite a simple process and does not need significant modifications to the model architecture or hyperparameters. It entails creating new training samples from existing data by using different transformations such as rotation, flipping, cropping, or introducing noise. This artificially expands the minority class, which helps to balance the class distribution. Second, data augmentation is data-driven, requiring no external knowledge or assumptions about the underlying data distribution. It makes use of the existing data to produce more varied instances, which may help the model generalize and make better predictions. The outcomes of augmentation are shown in Table 5 and Table 6. Overall process of data augmentation is shown in Algorithm 1.
**Algorithm 1:** Auto Augmentation**Steps****Explanation****Input****Output**Step 1Import the necessary libraries, including albumentations, torchvision, and torch.NoneNoneStep 2Define the function get_autoaugment_transform() to create the AutoAugment transformation pipeline. The function uses albumentations to specify the AutoAugment policy and other image transformations. The resulting transformation pipeline will resize the images, apply horizontal flip, and normalize the pixel values.AutoAugment policy defined using AlbumentationsNoneStep 3In the main part of the code:Load the custom dataset using ImageFolder from torchvision.datasets.Apply the previously defined AutoAugment transformation to the dataset.Custom dataset containing images organized by class labelsNoneStep 4Create a DataLoader using torch.utils.data.DataLoader. The DataLoader will handle batching and shuffling of the augmented dataset during training. Define a simple CNN model class SimpleCNN using nn.Module. Set uCustom dataset containing images organized by class labels-Augmented dataset-CNN architecture-Loss function-Optimizer-Hyperparameters-(Optional) Validation datasetNoneStep 5Define a simple CNN model class SimpleCNN using nn.Module. Set up the loss function (e.g., CrossEntropyLoss) and optimizer (e.g., SGD or Adam) for training the model. Specify the quantity of training cycles and other hyperparameters like learning rate and batch size.-CNN architecture-Loss function-Optimizer-HyperparametersNoneStep 6Train the model over the course of specified number of epochs:Set the model to training mode by calling model.train().Loop through the DataLoader to obtain batches of augmented images and their corresponding labels.To obtain expected outputs, run the model via a forward pass.Calculate the difference between ground truth labels and expected outputs using the defined loss function.Perform a backward pass to compute gradients of the model’s parameters with respect to the loss.Update the model’s parameters using the chosen optimizer and the computed gradients.-Augmented dataset-Model-Loss function-Optimizer-Hyperparameters-(Optional) Validation dataset-Trained CNN model-Model checkpoints (optional)-Training and validation performance metricsStep 7After training is complete, the model is ready for inference and evaluation on new data.-Trained CNN model-Inference dataInference results on new data

### 3.3. MixNet-LD Architecture

The MixNet-LD model is a deep neural network architecture created to effectively extract visual representations. MixNet-LD seeks a mix between performance and computational economy, drawing inspiration from the successful Inception and MobileNet models. Mix-depthwise convolutional blocks are used, which combine depthwise separable convolutions with different kernel sizes to enable effective feature extraction at various scales. Furthermore, the feature mixing of blocks with skip connections allows information to move easily across layers, improving gradient propagation. Convolutional layers, batch normalization, and activation functions are used in the model to extract features and account for nonlinearity. For further feature processing, fully linked layers are employed after global average pooling to minimize spatial dimensions. To obtain class probabilities for classification problems, a SoftMax activation function is applied in the ultimate stage as the output layer. Because of its emphasis on computational effectiveness, MixNet-LD is particularly suited for contexts with limited resources, such as mobile devices. The model gains superior performance in numerous computer vision tasks, such as picture classification and object recognition, by being trained on a huge dataset of labeled images. MixNet-LD has to be fine-tuned on a fundus dataset that has been suitably labeled in order to be used for identifying HR in retinography pictures. Below Figure 4, a schematic diagram is shown. The MixDepthwise Convolutional Block applies depthwise convolutions with different kernel sizes to the input feature map. Then, for each convolution, it follows the sequence of batch normalization (BN), ReLU activation, and pointwise convolution with a 1 × 1 kernel. Finally, the outputs obtained from all these convolutional paths are concatenated together.

Let x be the input feature map to the MixDepthwise Convolutional Block, K be the set of kernel sizes used in the block, Conv k (⋅) represent the depthwise convolution operation with a kernel size k, Conv 1x1 (⋅) represent the pointwise convolution operation with a 1×1 kernel. The MixDepthwise Convolutional Block can be represented as follows:(1)MixDepthwiseBlock(x)=Concatenate([conv1x1(ReLU(BN(convk(x))))]k∈K)

Equation (1) represents the MixDepthwise Convolutional Block. In this equation:(MixDepthwiseBlock(x)) is the output of the MixDepthwise Convolutional Block when applied to input\*x*).(Concatenate(cdot)) denotes concatenation along a certain dimension.([conv_1x_1(ReLU(BN(conv_k(x))))]_k∈K) represents a set of operations performed for each (k) in the set (K).(k) is an index that iterates over the set (K).(k) is used to indicate that the operations inside the MixDepthwise Convolutional Block are applied for each value of (k) in the set (K). This means that for each kernel size (k) in (K), the operations inside the square brackets are performed independently.(Concatenate(cdot)) is used to concatenate the results of these operations along a certain dimension. The specific dimension for concatenation might depend on the context of the implementation, typically along the channel dimension.The notation (k∈K) is used to indicate that (k) takes on values from the set (K). This is a common mathematical notation to express operations that are applied for each element in a set.(k∈K) is used to clarify that the operations inside the MixDepthwise Convolutional Block are applied for each kernel size (k) in the set (K), and (Concatenate(cdot)) is used to concatenate the results of these operations. The specific dimension for concatenation will depend on the implementation but is typically along the channel dimension.

Dense blocks, which are differentiated by their dense connectedness, are a fundamental component of the DenseNet design. Each layer inside a dense block is physically linked to every other layer, promoting efficient feature reuse by giving each layer access to the feature maps created by all previous levels. This not only encourages the extraction of fine-grained and high-level features, but it also alleviates the vanishing gradient issue, allowing for the training of extremely deep networks. Furthermore, dense blocks excel at parameter efficiency, lowering the amount of learnable parameters while preserving model expressiveness, making DenseNet an appealing alternative for image classification and other computer vision applications.

Dense blocks, which are included into the “MixNet-LD” system, are critical in enhancing the model’s categorization skills. Their extensive connectedness promotes dynamic information sharing across network levels, encouraging feature reuse and improving input data representation. This feature reuse not only allows the model to capture complicated patterns, but it also leads to more a robust gradient flow during training, hence mitigating the vanishing gradient issue that is common in deep neural networks. Furthermore, dense blocks excel in parameter efficiency, allowing the creation of deep models with fewer parameters than traditional designs while maintaining expressive capability.

The “MixNet-LD” system stands out as a powerful solution for image classification problems by integrating MixNet’s channel-level information flow with dense blocks’ feature reuse capabilities. It takes use of the synergy between these two architectural aspects to achieve cutting-edge precision while retaining computing economy. Dense blocks, in essence, are a critical innovation that improves the efficacy of the “MixNet-LD” system, providing a viable route for furthering the study of computer vision and picture categorization.

The dense block is a standard building block in DenseNet models, comprising two convolutions and a dense connection (Algorithm 2). Let x be the input feature map to the dense block, F be the number of filters (output channels) in the block, Conv(⋅) represent the convolution operation, shortcut be the identity shortcut (dense connection). The DenseBlock can be represented as follows:(2)DenseBlock(x,F)=ReLU(BN(conv(x,F))+shortcut)

The MixNet with Dense blocks can be represented as follows:$$x=Conv2D\left(inputs,64,\left(3,\;3\right),‘same’\right)$$
x=ReLU(BN(x))
x=MaxPooling2D(x,(2, 2))
x=MixDepthwiseBlock(x,K1)
x=MixDepthwiseBlock(x,K2)
x=DenseBlock(x,128)
x=DenseBlock(x,128)
x=Conv2D(x,256,(3,3),‘same’)
x=ReLU(BN(x))
x=GlobalAveragePooling2D(x)
x=Dense(x,256)=ReLU(BN(x))
x=Dense(x,N,SVM)
**Algorithm 2**: Working of the MixNet-DL model for feature map extraction**Steps****Explanation****Input****Output**Step 1Define the MixDepthwise Convolutional Block:Input: x (Input feature map), K (Set of kernel sizes used in the block)For each k in K:Perform a depthwise convolution on x with kernel size k.Apply Batch Normalization to the result. Apply ReLU activation to the result.Perform a pointwise convolution on the result with a 1×1 kernel.Concatenate the outputs of all the pointwise convolutions.Output: Concatenated tensor representing the MixDepthwise Convolutional Block output.x (Input feature map)K (Set of kernel sizes)Concatenated tensor representing the MixDepthwise Convolutional Block output.Step 2Define the Dense block:Input: x (Input feature map), F (Number of filters in the block)Save x as the shortcut for the residual connection.Perform a 3×3 convolution on x with F filters and ‘same’ padding.Apply Batch Normalization to the result.Apply ReLU activation to the result.Perform another 3×3 convolution on the result with F filters and ‘same’ padding. Apply Batch Normalization to the result. Add the result to the shortcut to create the residual connection. Apply ReLU activation to the result. Output: Output tensor representing the Dense block output.x (Input feature map)F (Number of filters)Output tensor representing the Dense block output.Step 3Define the MixNet with DenseNet model:Input: inputs (Input tensor to the model) Perform a 3×3 convolution on inputs with 64 filters and ‘same’ padding.Apply Batch Normalization to the result. Apply ReLU activation to the result.Perform MaxPooling with a 2×2 pool size on the result.Apply the MixDepthwise Convolutional Block on the result with 1 K 1 kernel sizes.Apply the MixDepthwise Convolutional Block on the result with 2 K 2 kernel sizes.Apply the Dense block on the result with 128 filters.Apply the Dense block on the result with 128 filters. Perform a 3 × 3 convolution on the result with 256 filters and ‘same’ padding.Apply Batch Normalization to the result. Apply ReLU activation to the result.Perform Global Average Pooling on the result to reduce spatial dimensions.Perform a Dense layer with 256 units on the result.Apply Batch Normalization to the result.Apply ReLU activation to the result. Perform a Dense layer with the number of classes (N) and SVM linear classifier activation for classification.Output: Output tensor representing the final classification probabilities.inputs (Input tensor to the model, e.g., image data)Output tensor representing the final classification probabilities.Step 4Example Usage:Input shape: (224,224,3) # Adjust input shape based on your data.Number of classes: N (e.g., 10 for image classification with 10 classes) Instantiate the MixNet with ResNet model using the defined architecture and provided input shape and number of classes.Compile the model using the proper metrics, loss function, and optimizer. for your specific classification task.Train the model on your image dataset.Estimate the simulation’s performing on a separate test dataset.Input shape: (224, 224, 3)Number of classes: NTrained and compiled MixNet-DL model ready for classification.

### 3.4. Recognition of Lungs Diseases

Automatically recognizing lung diseases from lung X-ray and CT scan images presents issues for CAD systems. The MixNet architecture is used to create the MixNet-LD method for effective picture classification in order to solve this. Figure 5 shows how skip links are put into the network to speed up learning. The name of the complete system is MixNet-LD. Multiple depthwise convolutional layers are followed by ReLU activation, max pooling, and batch normalization layers to generate the dense layers in MixNet’s dense blocks. To enable efficient training of the dense network, these layers are connected using skip connections. Three dense blocks (DBs) are used by MixNet-DL to efficiently construct trained three-CNN feature maps with substantial depth, resulting in dense blocks. Each DB retains the same input and output sizes throughout the feature learning phase. The feature maps are produced by a factor of two downsampling from each DB. This model makes use of three DBs. To carry out the categorization process, the network’s final fully linked output is enlarged with an extra layer. The dense learning sub-network’s properties are successfully divided into two categories by this layer. The number of nerve cells in this layer is maintained at 850 to guarantee peak performance. Batch normalization layers are included in MixNet as a pre-processing step, which enhances the training of deep neural networks and model convergence. The following is a description of the batch normalization formulae (Table 7):(3)$$B=\{\;X_{1\ldots m}\},\;\gamma ,\;\beta$$
(4)$$\left\{y_{i}={BN}_{\gamma \beta }\left(X_{i}\right)\right\}$$
(5)$$\mu_{B}\leftarrow \frac{1}{m}\;\sum_{i=1}^{m}Xi$$
(6)$$\sigma_{B}\leftarrow \;\frac{1}{m}\;\sum_{i=1}^{m}Xi-\mu_{B}$$
(7)$$X_{i}\leftarrow \;\frac{X_{i}-\mu_{B}}{\sigma_{B}^{2}+\epsilon }$$
(8)$$y_{i}\leftarrow YX_{i}+\beta ={BN}_{\gamma \beta }\left(X_{i}\right)$$
(9)$$DB\left(z\right)=Output-Input=a\left(z\right)-z$$
(10)$$a\left(z\right)=DB\left(z\right)+z$$

To improve its performance on job *a*(*z*), the entire dense block functions by analyzing the actual output. The layers are working hard to learn the residual, *DB*(*z*), according to a close inspection of Figure 4 because of the y-identity relationship. As a result, unlike traditional network layers that fully understand the output (*a*(*z*)), the layers in a dense network learn about the dense *DB*(*z*).

### 3.5. LSVM

The linear SVM machine learning classifier, with a train–test splitting ratio of 75% to 25%, is used for the classification job. This decision was made because of how well linear SVM performs overall and how well it handles tiny datasets. SVM is a classification approach that is well known for being superior to alternative classifiers and is commonly employed to solve practical issues. Instead of using conventional DL or machine learning classifiers, the authors chose to create a depth-wise separable CNN for computer vision or image classification problems. However, the linear SVM classifier was chosen for the current study because it is excellent at handling tiny datasets although not yet achieving well in high-dimensional environments. While using a linear layer in the context of an SVM classifier is unusual, it most likely serves a function within the larger architecture or experimental setting. SVMs have generally been linked with linear decision boundaries and margin optimization; nevertheless, this representation may use neural network architectures and deep learning frameworks for training and integration with other components. However, for a complete understanding of the technique, further information about why this specific representation was selected and how it affects the model’s performance, training process, or relationship with other architectural parts is required. Clarity in this regard would improve the paper’s contribution by revealing the logic behind this atypical strategy and its consequences for the research’s aims and findings.

Given that the task required binary categorization, this choice made sense. In the process to boost the efficiency of their approach and identify the most effective hyperplane to distinguish between abnormal and healthy cells in retinal images, the authors also used linear SVM. Typically, an LSVM takes a vector Y=(z1,I2,..., zn) and outputs a value y c Rn that may be written as:(11)out=(Weig,Ziv)+c

Weig stands for the weight in Equation (9), while *c* is the offset. Weig and c are real-valued parameters that were both learnt during training and are a part of the real numbers (R) set. Dependent on whether y is larger than or less than 0, the input vector Ziv is classified as belonging to class 1 or class 2, as shown in Algorithm 3.
**Algorithm 3**: Proposed LSVM classifier.**Steps****Explanation**InputExtracted feature map x=(z1, z2,... , zn) with observations x=0,1, Test data ZtestOutputCategorization of normal and abnormal sections1Primarily, the classifier and kernelRegularize L2 parameters are labeled for optimization.2Creation of LSVM
**a**. The training procedure of LSVM is finished. using obtained attributes x=(z1, z2,…, zn) by our Algorithm 2. 
**b**. For the production of the hyperplane, use Equation (6).3The class label is assigned fortesting samples
ztest using the evaluation work of the equation further down.
                   Ztest=(Weig, Ziv)+c


## 4. Results

A collection of 13,313 lung pictures, including both high-resolution normal and diseased images, was used to train the MixNet-DL model. These lung photos were obtained from credible Pakistani hospitals (Pak-Lungs) and a number of reliable online sources. All 13,313 images were downsized to pixels for feature extraction and classification activities. Utilizing MixNet and leftover building blocks, the MixNet-DL system underwent training for 100 epochs. The best model was found in the 30th century and achieved an excellent F1-score of 0.97. The accuracy (ACC), specificity (SP), and sensitivity (SE) values for the proposed MixNet-DL system were determined by statistical analysis. Then, these measurements were contrasted with those from other systems. The MixNet-LD system was designed and built on a computer with an HP-i7 CPU with 8 cores, 32 GB of RAM, and a 2 GB Gigabyte NVIDIA GPU. Windows 11 Professional 64-bit is the operating system installed on this machine.

### 4.1. Experiment 1

In order to estimate the execution of the DL models VGG16, VGG19, InceptionV3, ResNet, Xception, and MobileNet with the proposed MixNet-DL system, we ran an experiment in this study. Notably, the same number of epochs was used to train each of these DL models. Two identical deep neural networks were trained after the top network was determined based on validation accuracy. Table 8 exhibits the comparison findings between the MixNet-DL system and the VGG16, VGG19, InceptionV3, ResNet, Xception, MobileNet, and Densenet-169 models in terms of sensitivity, specificity, accuracy, and area under the curve (AUC). According to the results, the MixNet-LD system performs better than other DL models, proving its superior performance. Figure 6 represents the comparison between different DL models and MixNet-LD.

### 4.2. Experiment 2

In this study, we use a dataset called “lung disease dataset (four types)” which we downloaded from a reliable online source [44], to evaluate our performance suggested MixNet-DL approach. In the beginning, we used the appropriate datasets to compare the model’s performance on the training and validation sets and to assess the loss function. The training and validation accuracy of the MixNet-DL model as it was trained on this dataset is shown visually in Figure 7 and Figure 8. The outcomes unequivocally show how effectively our model works in both the training and validation settings. Furthermore, we obtained perfect accuracy on both the training and validation sets using the lungs dataset mentioned in Table 9.

### 4.3. Experiment 3

In this investigation, we estimated the effectiveness of our suggested. MixNet-DL method using the Pak-Lungs dataset. First, we evaluated the loss function and the representation’s operation on the training and validation sets using the Pak-Lungs dataset. The confusion matrix and training and validation accuracy of the MixNet-DL model when trained on this dataset are shown graphically in Figure 9 and Figure 10. The outcomes show our model’s great efficacy in both training and validation settings. Additionally, we achieved a 99% accuracy rate on both the training and validation sets by using the Pak-Lungs dataset mentioned in Table 9.

### 4.4. Experiment 4

In this paper, we test the efficacy of our proposed MixNet-LD technique using a novel dataset named Pak-Lungs (cancer) and Chest CT-Scan images dataset [48], which was collected from hospitals in Pakistan. We began by comparing the performance of the model on both training and validation datasets, as well as assessing the loss function on the corresponding datasets. The accuracy of the MixNet-LD model during training and validation using this dataset is depicted in Figure 11 and Figure 12. The findings show that our model performed exceptionally well in both the training and validation settings. Using the Pak-Lungs dataset, we achieved good accuracy on both the training and validation sets. Additionally, we were able to obtain 98.2% accuracy on both the training and validation sets by using the Pak-Lungs (cancer) dataset and 99% accuracy on the Chest CT-Scan images dataset [48]. Both comparisons are shown in Figure 11 and Table 10.

### 4.5. Experiment 5

Table 11 presents a detailed performance comparison between the proposed MixNet-LD and other existing architectures, such as ResNet50, MobileNet, and D-Resnet, for the classification of various lung disease classes including normal, COVID-19, pneumonia, and tuberculosis. As demonstrated in the table, ResNet50 [19] shows sensitivity, specificity, F1-score, recall, and accuracy scores of 0.77, 0.81, 82, 0.81, and 82.10, respectively. These scores, while substantial, are significantly outperformed by the succeeding architectures.

MobileNet [20] posts improvements in all metrics, achieving a sensitivity of 0.82, specificity of 0.83, F1-score of 84, recall of 0.85, and an accuracy of 84.55. Further enhancement is noted in the D-Resnet [21] results, with scores of 0.84, 0.85, 87, 0.86, and 85.20 in sensitivity, specificity, F1-score, recall, and accuracy respectively.

However, the proposed MixNet-LD model stands out with remarkable superiority, recording near-perfect scores across all metrics. The model demonstrates a sensitivity and recall of 0.99, specificity of 0.985, an F1-score of 0.988, and an astonishing accuracy of 0.99. This significant outperformance of MixNet-LD underscores its advanced capabilities and effectiveness in accurately classifying diverse lung conditions, solidifying its position as a leading tool in the field, as per the presented data.

## 5. Discussion

Several lung diseases, including pneumonia, tuberculosis (TB), COVID-19, and lung cancer, may now be detected and classified thanks to substantial breakthroughs achieved by DL in medical image analysis. With high morbidity and mortality rates, these illnesses represent serious problems for public health internationally. For rapid treatment and better patient outcomes, diagnosing and classifying these illnesses as early as possible is essential. CNNs have demonstrated excellent performance in diagnosing lung illnesses in recent years. DL models for analyzing chest X-rays have been developed partly due to research that introduced the ChestX-ray8 dataset. The basis for identifying pneumonia and other thoracic illnesses has been established using this dataset in conjunction with weakly supervised classification and localization approaches. For effective picture classification, multi-scale dense networks utilizing deep CNN architectures should be introduced. By extracting hierarchical characteristics, this method has proven effective in reliably detecting lung illnesses.

The COVID-19 pandemic’s advent presented a necessity for accurate and speedy detection. An artificial intelligence system can distinguish COVID-19 from community-acquired pneumonia in chest CT images. This method helps identify COVID-19 patients by utilizing DL techniques, which improve disease treatment and control. A DL model for automatic pulmonary TB categorization on chest radiography has been tested. This approach enabled rapid screening and diagnosis by precisely identifying TB-related abnormalities. DL’s application of the COVID-19 diagnosis has drawn a lot of interest for transfer learning and CNNs in placing COVID-19 in X-ray pictures. This method showed how DL may help radiologists and other healthcare professionals swiftly identify COVID-19 patients. Focusing on interstitial lung illnesses, we developed a deep CNN for lung pattern classification. In addition, we developed and evaluated an AI system for COVID-19 diagnosis that correctly identified COVID-19 patients from imaging data using DL techniques.

Along with infectious diseases, the diagnostic potential of DL models for lung cancer has also been thoroughly investigated. An image-based DL system can identify medical diagnoses, such as lung cancer. This model illustrated how DL may be used to detect the planned MixNet-LD system for the automatic identification and classification of lung diseases. The lungs are essential for the respiratory system, and they must function properly for the body to exchange oxygen and carbon dioxide. However, several illnesses can impact the lungs and have adverse health effects. Since it might be challenging to identify and extract the lesion characteristics connected to lung problems, early and precise diagnosis of lung diseases is crucial for prompt and effective therapy. To solve this problem, the MixNet-LD system combines the strength of MixNet, a particular pre-trained model, with dense blocks. By minimizing noise and emphasizing abnormalities, the suggested pre-processing technique utilizing Grad-Cam improves the quality of pictures of lung illness, further enhancing the classification’s performance. Data augmentation techniques are also used to correct the dataset’s class imbalance and avoid overfitting, strengthening, and improving the MixNet-LD model. The creation of the Pak-Lungs dataset, which consists of 6000 photos of lung diseases gathered from reliable online sources and Pakistani hospitals, is a significant addition to the study. The MixNet-LD model successfully handled various lung illness cases thanks to this sizable dataset, achieving a fantastic accuracy of 98.5% on the challenging lung disease dataset. Further demonstrating the superiority of the MixNet-LD system, which exceeds previous techniques with a substantially higher accuracy of 95%, is the comparison with state-of-the-art methods in medical image processing. As it offers a trustworthy and automated way of correctly diagnosing normal COVID-19, pneumonia, tuberculosis, and lung cancer illnesses, the system’s performance can potentially have real-world ramifications in medical diagnosis and therapy. Figure 13 represents the lung patterns detected by the MixNet-LD system and Figure 14 shows the visual results of the predicted MixNet-LD system.

The MixNet-LD system successfully combines MixNet with dense blocks and uses data augmentation methods. While MixNet’s pre-trained model offers the benefit of transfer learning, allowing the system to learn from past knowledge and attain cutting-edge performance, the dense blocks will enable the extraction of critical characteristics from the pictures. Using data augmentation approaches makes the model more flexible and able to handle various lung illness situations while lowering the danger of overfitting.

Overall, the MixNet-LD method better presented state-of-the-art methods and performed well, making it an essential tool in medical image analysis. It can also handle a variety of lung disease scenarios. The MixNet-LD system can dramatically boost patient outcomes and healthcare management by delivering accurate and timely diagnostics of lung illnesses. Further investigation can look at the possibilities of the MixNet-LD system in various clinical contexts and the applicability of the technology to other medical image analysis jobs. The system’s breadth and influence in medical imaging may be increased by ongoing improvement and optimization, ultimately leading to more efficient medical image processing in clinical settings. Figure 13 demonstrates the compatibility of the m-Xception system with X-ray pictures, hence highlighting its potential benefits for significant biomedical applications. The illness-identified regions within images exhibit a higher level of effectiveness in the context of computer-aided diagnosis (CAD) systems, as they contribute to a streamlined process of disease classification when utilizing chest X-ray images.

The MixNet-LD system has far-reaching and varied implications for medical image processing in clinical settings. MixNet-LD has the potential to transform how medical pictures are evaluated, interpreted, and used in healthcare by using the power of deep learning. Its significance in improving diagnostic accuracy is especially notable since it may aid healthcare workers in accurately detecting disease disorders. This results in not just early interventions but also more personalized treatment strategies, eventually improving patient outcomes. Furthermore, MixNet-LD improves operational efficiency in clinical processes. It can scan and triage a large amount of medical pictures quickly, prioritizing instances that need urgent treatment. This is particularly useful in resource-constrained healthcare settings where rapid decision making is crucial. MixNet-LD may also relieve the strain on radiologists and medical personnel by automating typical image analysis operations, enabling them to concentrate on complicated situations and provide more complete patient care. The model’s consistency in picture interpretation, as well as its potential for telemedicine applications, highlight its practical importance in current healthcare delivery. However, like with any new technology in clinical practice, extensive validation, ethical concerns, and continual refinement are required to guarantee MixNet-LD’s responsible and successful incorporation into the medical environment.

The research around the MixNet-LD system has the potential to profoundly transform real-world healthcare procedures. This technology promises to improve diagnosis accuracy, expedite resource allocation, and minimize the strain of healthcare personnel by using deep learning capabilities. Patients will benefit from earlier and more exact diagnoses, which will lead to earlier treatments and better treatment results. Furthermore, MixNet-LD’s function in telemedicine and clinical decision support may broaden access to healthcare and provide vital insights for personalized treatment. This research has the potential to expedite the training of healthcare professionals and accelerate the pace of discovery, ultimately contributing to a patient-centric healthcare ecosystem that is more efficient, effective, and responsive to the evolving needs of both patients and providers. However, careful consideration of ethical, privacy, and security issues will be required to enable MixNet-LD’s responsible and successful incorporation into real-world healthcare procedures.

### 5.1. Advantages of Proposed System

The paper concludes by highlighting the significant strides made by the MixNet-LD approach in the realm of automated lung disease severity classification. It accentuates the critical need for accurately identifying and categorizing the severity of lung diseases, as this information forms the cornerstone of effective medical diagnosis and treatment strategies. The MixNet-LD methodology, founded upon an enhanced pre-trained MixNet model, emerges as a promising solution poised to tackle this complex challenge:(1)At the heart of the conclusion lies the remarkable performance of the MixNet-LD system in precisely categorizing the severity of lung diseases. The approach achieves a level of accuracy and learning capabilities that position it at the forefront of advancements in the field. This achievement underlines the potential of the approach to significantly enhance the accuracy of classification models within the domain of medical image analysis, a critical aspect of modern healthcare diagnostics.(2)Integral to the success of the MixNet-LD approach are the adept utilization of data augmentation strategies and pre-processing techniques, such as Grad-Cam. These methods not only improve the quality of the input data but also improve the classification performance of the model. Furthermore, the paper underscores the importance of addressing the inherent challenge of unbalanced class distributions in the dataset. By incorporating dense blocks and data augmentation techniques, the MixNet-LD approach effectively navigates this challenge, ensuring that the model’s performance remains robust across all severity categories.(3)A notable aspect of the MixNet-LD system is its ability to achieve remarkable results without sacrificing model size and complexity. This pragmatic approach ensures that the model’s deployment remains feasible in real-world clinical settings, where resource constraints and practical considerations are paramount. By striking a balance between performance and complexity, the MixNet-LD approach attains a degree of practicality that enhances its potential for widespread adoption and implementation.(4)The generalizability of the MixNet-LD approach is another key takeaway from the conclusion. The approach is put to the test across a diverse array of datasets, including both a publicly available and novel private dataset known as Pak-Lungs. This comprehensive evaluation underscores the adaptability and effectiveness of the MixNet-LD approach, suggesting its potential applicability in various clinical contexts.(5)By advancing the accuracy of automated disease severity classification, the MixNet-LD system adds a noteworthy contribution to the broader field of medical image processing. This contribution resonates beyond the confines of the paper, suggesting promising directions for the development of more effective strategies for medical diagnosis and treatment.

Overall, the advantage solidifies the MixNet-LD approach as a powerful tool in automating the classification of lung disease severity, with far-reaching implications for the advancement of medical image analysis within clinical settings.

### 5.2. Challenges and Future Works

Descriptions of potential future works that could build upon the achievements of the MixNet-LD approach in the field of automated lung disease severity classification are given as follows:Future research could explore the integration of multiple modalities of medical imaging data, such as CT scans and X-rays, into the MixNet-LD approach. Combining information from diverse sources could enhance the model’s ability to accurately classify disease severity and provide a more comprehensive diagnostic tool.Developing methods to interpret and explain the decisions made by the MixNet-LD model could enhance its clinical utility. Research could focus on creating visualization techniques that highlight the regions of interest in medical images that contribute to the severity classification, aiding medical professionals in understanding and trusting the model’s predictions.While MixNet-LD has shown promise in lung disease severity classification, similar approaches could be investigated for other medical conditions. Researchers could adapt and extend the methodology to automate severity assessment in diseases affecting different organs or systems, contributing to a wider range of clinical applications.Exploring techniques to deploy the MixNet-LD approach in real-time diagnosis could revolutionize patient care. Developing optimized models that can provide severity assessments quickly and accurately would be valuable for guiding immediate medical decisions and interventions.Conducting large-scale clinical studies to validate the MixNet-LD approach’s performance on diverse patient populations would bolster its credibility and real-world applicability. Collaborations with medical institutions could provide access to extensive datasets, allowing researchers to assess the model’s performance across various demographics.Integrating the MixNet-LD approach into clinical decision support systems could provide physicians with an additional layer of information when making treatment recommendations. The model’s predictions could be used as a complementary tool to aid doctors in diagnosing and planning patient care.Developing mechanisms for online learning and continuous model improvement could allow the MixNet-LD approach to adapt to emerging patterns and changes in lung disease severity. The model could be trained on newly available data to ensure its relevance and accuracy over time.Investigating the potential benefits of ensemble methods, where multiple models are combined, could enhance the overall performance of the MixNet-LD approach. Combining different architectures or training strategies could lead to improved accuracy and robustness.In addition to other DL approaches, NASNet, MobileNet, and EfficientNet model analysis of fresh datasets may be performed in the future.

In summary, the future of research in this area holds promising opportunities for further innovation and impact. Continued exploration and refinement of the MixNet-LD approach, along with addressing challenges and expanding its scope, could lead to substantial advancements in automated lung disease severity classification and broader applications in medical image analysis.

## 6. Conclusions

In this study, a deep learning-based architecture was developed and utilized to classify chest illnesses into several categories, including normal, COVID-19, pneumonia, tuberculosis, and lung cancer. The architecture was trained and evaluated using chest X-ray images. The MixNet-LD system, as detailed in the paper, emerges as a pioneering solution in the precise classification of diverse lung diseases, achieving an exceptional 98.5% accuracy rate. The system leverages a robust pre-processing strategy, employing Grad-Cam and data augmentation to enhance its performance and avoid overfitting, even with imbalanced class distributions. The use of dense blocks further refines its classification capabilities across various lung disorders. Tested on multiple datasets, including a novel “Pak-Lungs” dataset, the model’s consistent high performance underscores its reliability and effectiveness. The innovative design of the MixNet-LD system, featuring additional layers and effective feature extraction using CNN, marks a significant advancement in medical image processing. This research highlights the potential of the MixNet-LD system to greatly improve the accuracy of lung disease classification, offering a promising tool for enhanced clinical diagnosis and treatment strategies.

## Figures and Tables

**Figure 1 diagnostics-13-03195-f001:**
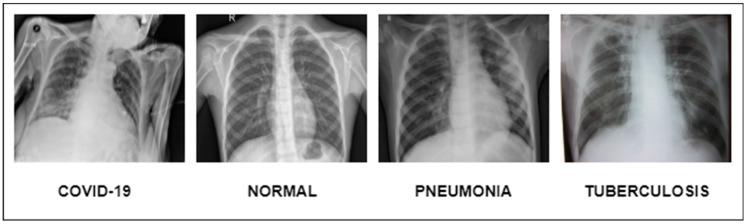
Illustration of lung diseases.

**Figure 2 diagnostics-13-03195-f002:**
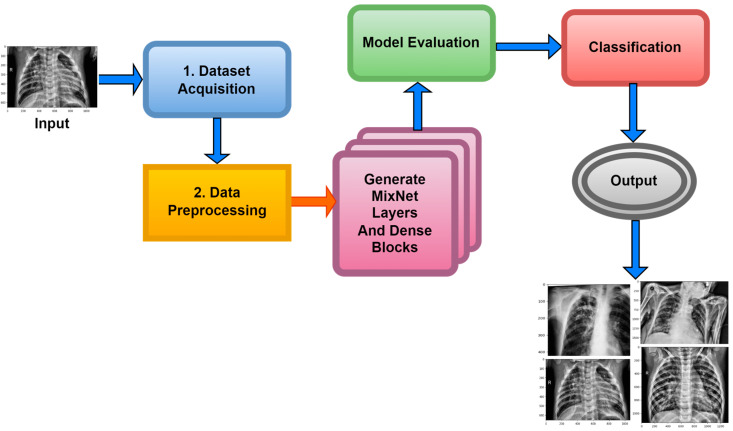
The MixNet-LD system’s well-organized flow diagram for identifying lung illnesses.

**Figure 3 diagnostics-13-03195-f003:**
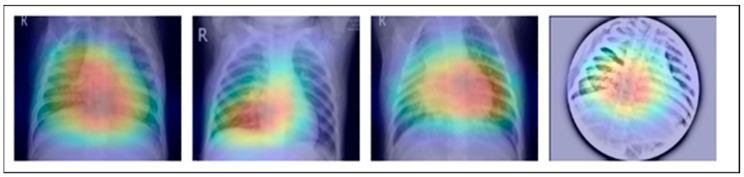
This image represents pre-processing results after the Grad-Cam technique.

**Figure 4 diagnostics-13-03195-f004:**
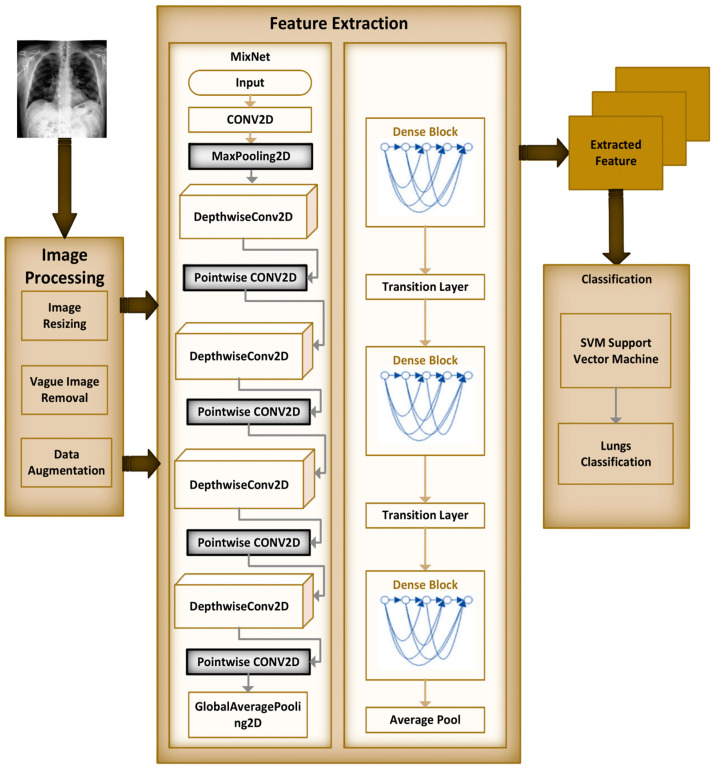
Schematic diagram of MixNet-LD.

**Figure 5 diagnostics-13-03195-f005:**
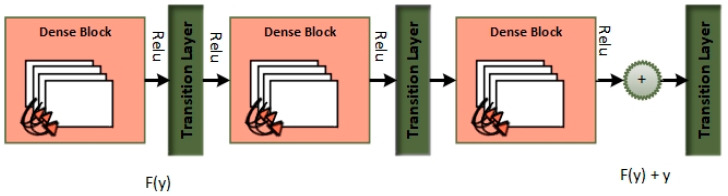
Dense blocks.

**Figure 6 diagnostics-13-03195-f006:**
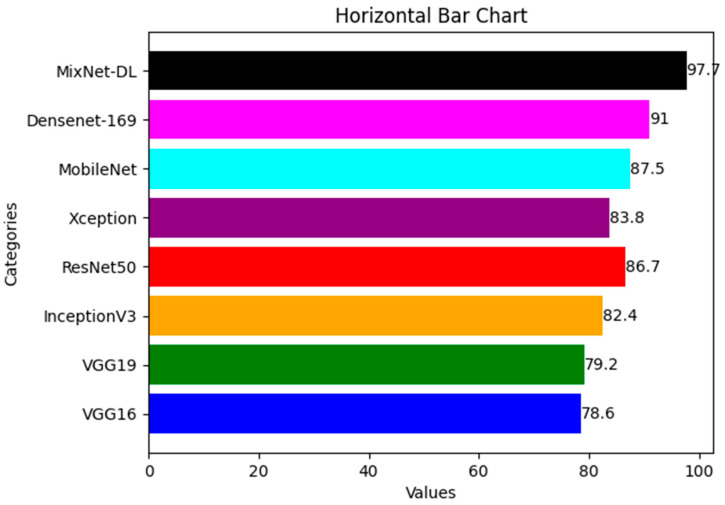
Comparison between different DL model and MixNet-LD.

**Figure 7 diagnostics-13-03195-f007:**
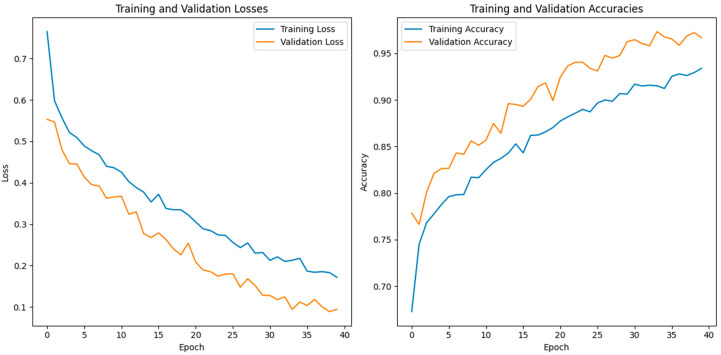
The proposed model’s training validation accuracy and loss.

**Figure 8 diagnostics-13-03195-f008:**
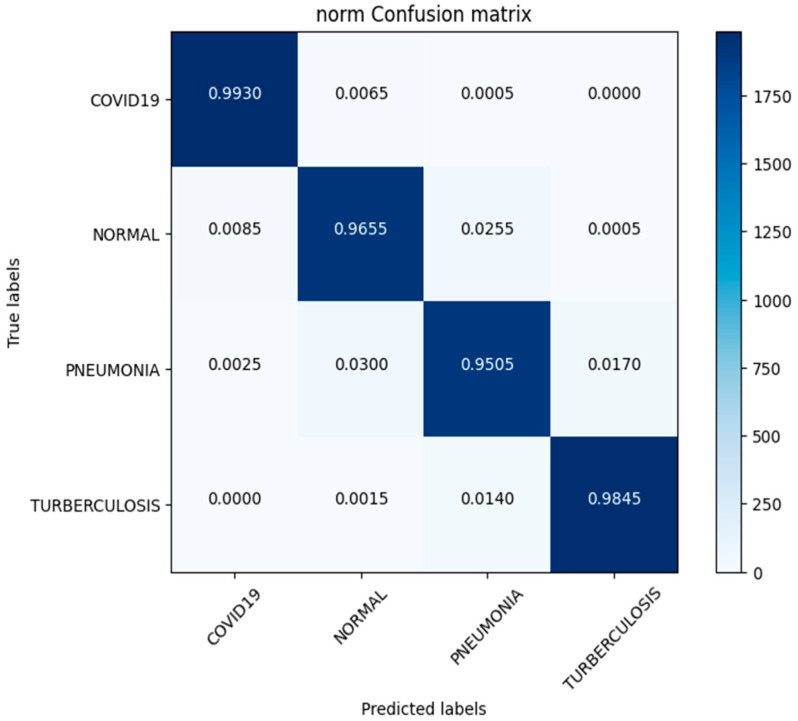
Confusion matrix of lung disease dataset (4 types).

**Figure 9 diagnostics-13-03195-f009:**
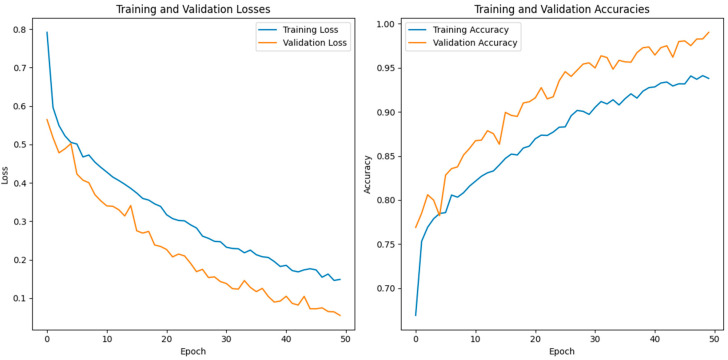
Illustration of the training and validation accuracy and loss of the proposed model using Pak-Lungs.

**Figure 10 diagnostics-13-03195-f010:**
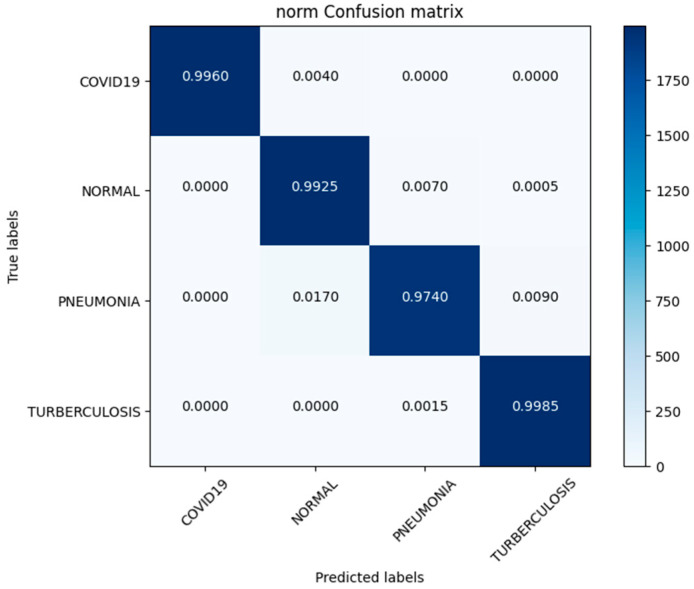
Confusion matrix of Pak-Lungs dataset.

**Figure 11 diagnostics-13-03195-f011:**
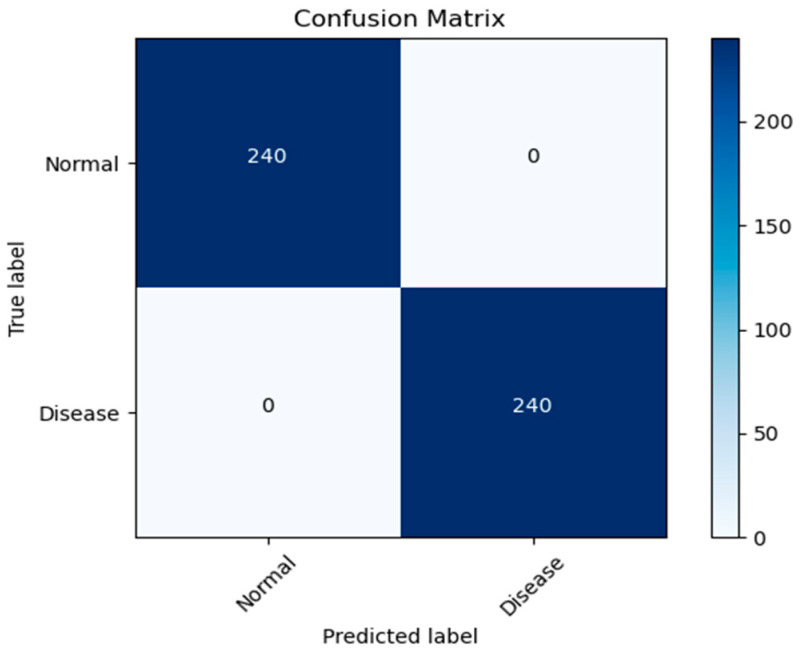
Confusion matrix of Pak-Lungs (cancer).

**Figure 12 diagnostics-13-03195-f012:**
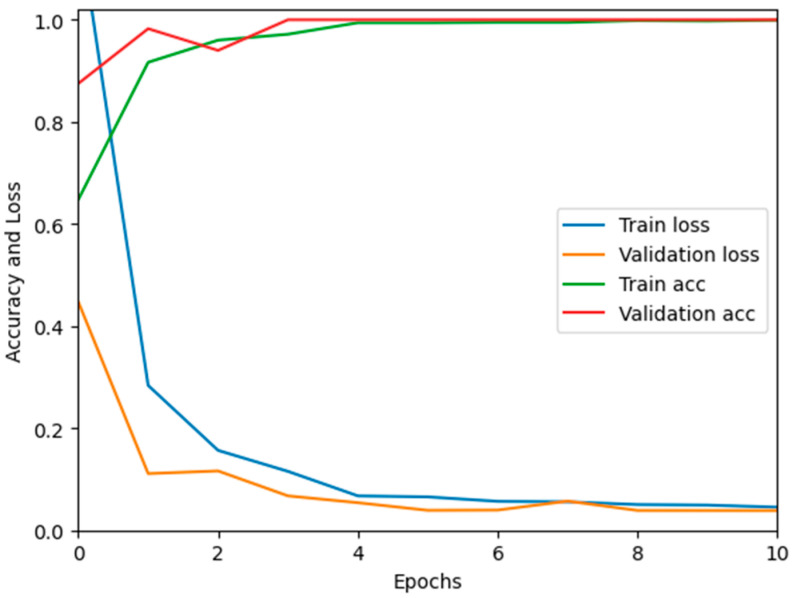
Training validation accuracy and loss on Pak-Lungs (cancer).

**Figure 13 diagnostics-13-03195-f013:**
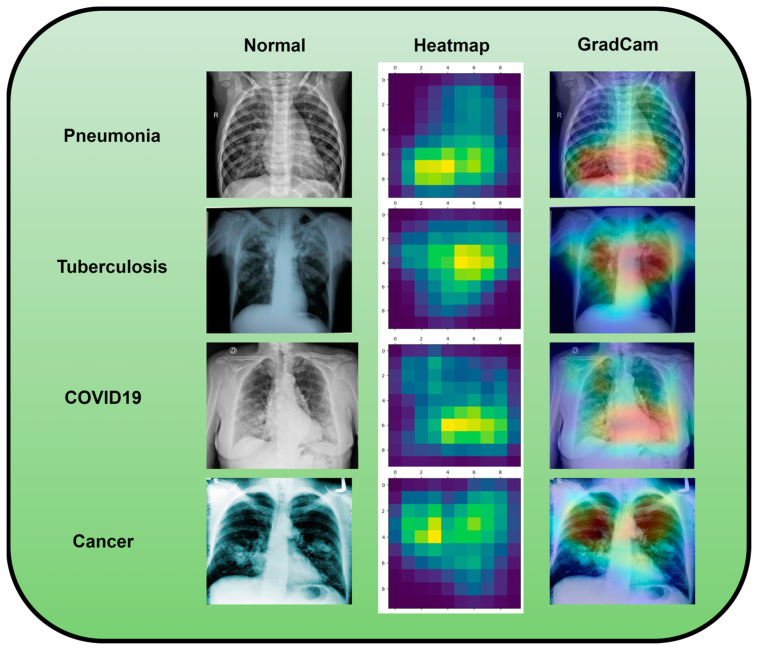
An investigation of the proposed MixNet-LD system’s decision capabilities based on chest X-ray disorders (pneumonia, tuberculosis, COVID-19, cancer, and normal) to highlight patterns using the Grad-Cam technique.

**Figure 14 diagnostics-13-03195-f014:**
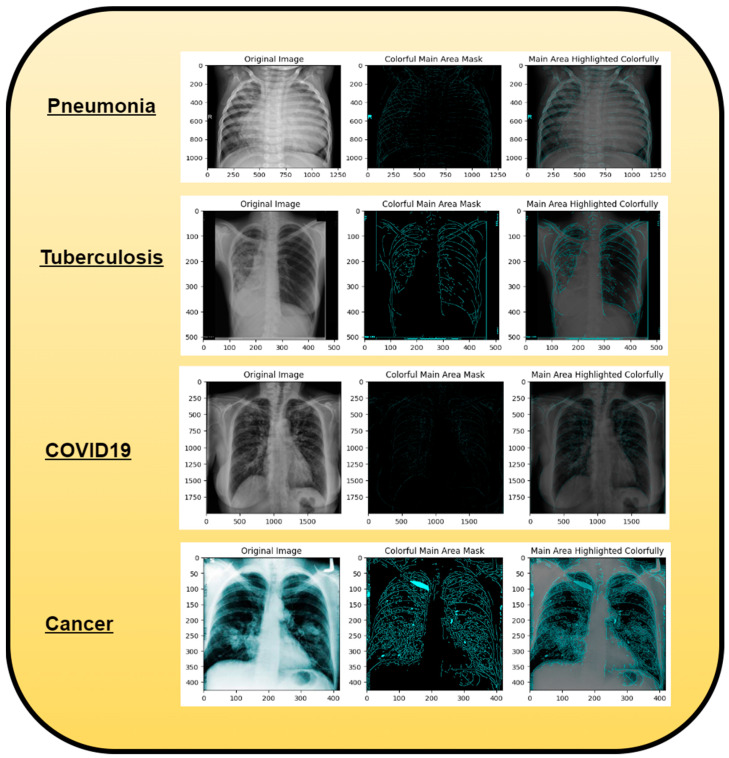
A graphical representation illustrating the anticipated outcomes generated by the proposed MixNet-LD paradigm.

**Table 3 diagnostics-13-03195-t003:** Dataset of lung diseases for the MixNet-LD framework.

Ref	Datasets	Normal	COVID-19	Pneumonia	Tuberculosis	Total
[44]	Lung diseases (4 types)	1342	462	3872	660	6336
Private	Pak-Lungs	1500	1500	1500	1500	6000
		**2842**	**1962**	**5372**	**2160**	**12,336**

**Table 4 diagnostics-13-03195-t004:** Dataset of lung cancer for the MixNet-LD framework.

Ref	Dataset	Normal	Cancer	Total
[52]	Chest CT-Scan	154	473	627
Private	Pak-Lungs	175	175	350
		**329**	**648**	**977**

**Table 5 diagnostics-13-03195-t005:** Image dataset of the lung cancer for the MixNet-LD system.

Normal	Cancer	Total Images
500	500	1000
500	500	1000
**1000**	**1000**	**2000**

**Table 6 diagnostics-13-03195-t006:** Image dataset of lung diseases for the MixNet-LD system.

Normal	COVID-19	Pneumonia	Tuberculosis	Total Images
2000	1500	4000	1500	9000
2000	1500	4000	1500	8000
**4000**	**3000**	**8000**	**3000**	**17,000**

**Table 7 diagnostics-13-03195-t007:** Symbolization table.

Methods	Values
B Batch	Batch
x batch minimum activating value	Batch minimum activating value
$\mu_{B}$ mini-batch mean	Mini-batch mean
$\sigma_{B}^{2}$	Mini-batch variance
ϵ	Adding a constant to ensure numerical stability
β	Learning variable
γ	Learning variable

**Table 8 diagnostics-13-03195-t008:** Performance comparison between MixNet-LD, VGG16, VGG19, InceptionV3, ResNet, Xception, and MobileNet.

Models	Sensitivity %	Specificity %	Accuracy %
VGG16	72	76	78.6
VGG19	75	77	79.2
InceptionV3	80.1	81.5	82.4
ResNet50	82.5	84.9	86.7
Xception	79.2	80.4	83.8
MobileNet	83.6	84.7	87.5
Densenet-169	91	91	91
**MixNet-DL**	**95.2**	**96.7**	**97.7**

**Table 9 diagnostics-13-03195-t009:** Performance assessment between Lung Disease Dataset (4 Types) and Pak-Lungs.

Dataset	Sensitivity	Specificity	F1-Score	Recall	Accuracy
Lung disease dataset (4 types) [51]	0.98	0.97	98	0.98	97
Pak-Lungs	0.99	0.98	0.98	0.99	0.99

**Table 10 diagnostics-13-03195-t010:** Performance comparison between Chest CT-Scan images dataset and Pak-Lungs (cancer).

Dataset	Sensitivity	Specificity	F1-Score	Recall	Accuracy
Chest CT-Scan images dataset [52]	0.98	0.99	99	0.99	99
Pak-Lungs (cancer)	0.98	0.99	0.98	0.99	0.99

**Table 11 diagnostics-13-03195-t011:** State-of-the-art performance comparison between MixNet-LD and other architectures using various classes such as normal, COVID-19, pneumonia, and turberculosis.

Methods	Sensitivity	Specificity	F1-Score	Recall	Accuracy
ResNet50 [19]	0.77	0.81	82	0.81	0.82
MobileNet [20]	0.82	0.83	84	0.85	0.84
D-Resnet [21]	0.84	0.85	87	0.86	0.85
Proposed MixNet-LD	0.99	0.98	0.98	0.99	0.99

## Data Availability

For experimental purposes, one private dataset named Pak-Lungs was utilized. Other publicly available datasets were retrieved on from https://www.kaggle.com/datasets/omkarmanohardalvi/lungs-disease-dataset-4-types (accessed on 28 July 2023) and the “Chest CT-Scan Images Dataset” was retrieved from www.kaggle.com, www.kaggle.com/datasets/mohamedhanyyy/chest-ctscan-images, accessed on 28 July 2023.

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
