# Peer review of "MixNet-LD: An Automated Classification System for Multiple Lung Diseases Using Modified MixNet Model"

_diagnostics, 2023, doi:10.3390/diagnostics13203195_

Round 1
Reviewer 1 Report
The article presents MixNet-LD, an automated classification system designed for the categorization of lung diseases using a modified pretrained MixNet model. Through preprocessing techniques like Grad-Cam, data augmentation, and dense blocks, the system aims to enhance the classification of different severities of lung conditions. Tested across various datasets, including a novel one named "Pak-Lungs," the MixNet-LD model purportedly achieves a high accuracy rate of 98.5%. The research highlights its potential in improving medical image analysis, emphasizing its performance while maintaining practical model size and complexity, suitable for real-world clinical applications. The paper needs a major revision to address all critical issues as outlined below:
· The paper mentions datasets "gathered from credible internet sources," which is concerning due to its vagueness. A rigorous scientific paper should provide detailed information about the datasets, including their origins and the methods used to curate them.
· The literature review missed some recent studies, the discussion thereof could improve this part of the paper: Sofia et al. (2023). A Neutrosophic Set Approach on Chest X-rays for Automatic Lung Infection Detection. Khan et al. (2021). VGG19 network assisted joint segmentation and classification of lung nodules in CT images. Jaszcz et al. (2022). Lung X-Ray Image Segmentation Using Heuristic Red Fox Optimization Algorithm. Rehman et al. (2021). A self-activated cnn approach for multi-class chest-related covid-19 detection.
· The introduction of the MixNet-LD system seems promising, but the abstract does not clearly establish what makes it significantly novel from other existing methods. The distinction between the "upgraded pretrained MixNet model" and the original MixNet model isn't clear either.
· The paper claims an impressive accuracy of 98.5% on a "difficult" lung disease dataset. However, the paper does not clearly outline the conditions under which this performance was achieved or provide comparisons with existing benchmarks.
· The paper mentions the utilization of a "preprocessing strategy that uses Grad-Cam," but it doesn't provide details about how Grad-CAM was used or why it was chosen over other AI explainability methods.
· The paper acknowledges the issue of unbalanced class distributions, but it fails to offer a comprehensive solution beyond data augmentation strategies. It's essential to explore more advanced techniques to handle class imbalance for more consistent results.
· The paper briefly discusses using a linear layer of the SVM classifier. However, it's unusual to describe SVM in terms of linear layers and activation functions. Further clarity on the exact architecture and why it was chosen is necessary.
· Although the paper boasts about the generalizability of MixNet-LD, testing on a novel private dataset ("Pak-Lungs") without providing details regarding its size, distribution, or collection methods can raise concerns.
· The conclusion boasts extremely high performance metrics across the board (sensitivity, specificity, F1-Score, recall, accuracy, and AUC). Such uniformly high metrics are suspicious, especially without a detailed discussion of the validation approach or potential sources of bias.
· The statement suggesting that "fusing CXR and CT images may be a potential advancement for future studies" lacks foundational evidence in the paper. While the idea is intriguing, it's presented without clear justification.
· The summary and conclusion sections contain significant overlaps in the information they provide. A more concise and distinct separation between the main findings and overall implications would improve clarity and impact.
In summary, while the MixNet-LD system promises a breakthrough in lung disease classification, the paper lacks depth in explaining its methodology, datasets, and validation approach. The high performance claims need rigorous validation and comparison with existing benchmarks to gain credibility. Detailed descriptions, clear differentiation from existing methods, and a thorough analysis of results are essential for the paper to be taken seriously in the scientific community.
Author Response
Dear Reviewer,
In the attachment, you can find the detailed response letter to your valuable comments. Thank you
Best regards,

Reviewer 2 Report
This paper presents MixNet-LD, an automated method using an upgraded MixNet model to classify lung disease severity.
Strength
- Created a new set of data, PAK-LUNGS, for Lung disease detection.
- Proposed MixNet-LD system.
Weakness
- Proposed MixNet-LD system is a combination of existing ideas(MobileNet + DenseNet) for lung disease classification.
(major)
- From "Applied GrandCam technique for image pre-processing to clear the images features and eliminate the interference in the image shown in figure 2." -> Before using Grad-CAM, the model should be fitted. What model is used for Grad-CAM result generation? If you used the same MixNet-LD model, then why don't you put the results after the model explanation?
-
(abstract)
- From "... to build a consistent an automated approach for ..." should be "... to build a consistent and automated approach for ..."?
(Introduction)
- From "DL algorithms (DLAs)... " The abbreviation, DL, should be written in full. DLA appears in the discussion section but I am not sure whether this abbreviation is necessary. I think "Deep Learning (DL) algorithms" should be enough.
- No explanation about Figure 1 in the introduction section, and it would be nice to have the name of the disease for each picture. Figure 1 is referenced with Figure 3, it may be better to combine these two figures together with and without the application of GradCAM. Also, it would be nice to add a disease label for each picture.
(3.3. MixNet-LD Architecture)
- From "... inspiration from the successful Inception and MobileNet models." -> Cite MobileNet and InceptionNet papers.
- Line 433-437: It is hard to understand these lines. Please rewrite mathematically. For example in Equation (1) ???????â„Ž?????????(?)=???????????([????1?1(????(??(?????(?))))]?∈?) (1) ,
why you describe ?∈? there? ??????????? is for what dimension? Notations are hard to understand.
- From Figure 4, I think dark conv2D box is 1by1 pointwise convolution just like MobileNet, specifying Conv2D (1by1) or specifying pointwiseConv2D would be nicer?
- DenseNet paper is also missing. cite it, please.
(4.1) Experiment 1
- Authors compared their results with VGG16, VGG19, InceptionV3, ResNet, Xception, and MobileNet models. However, they are not lung disease-specific models. if there are open-source implementations of other LD prediction models, including one or two of them would be even nicer.
Author Response

(The authors gave the same response as above.)

Reviewer 3 Report
The article introduces MixNet-LD, an automated approach utilizing a pretrained MixNet model to categorize lung disease severity. It employs Grad-Cam preprocessing and data augmentation to enhance classification.
1- What is the primary objective of the MixNet-LD system described?
2- It is suggested to add the following reference in the introduction:-
- Sergio Saponara, Abdussalam Elhanashi, Alessio Gagliardi, "Reconstruct fingerprint images using deep learning and sparse autoencoder algorithms," Proc. SPIE 11736, Real-Time Image Processing and Deep Learning 2021, 1173603 (12 April 2021); https://doi.org/10.1117/12.2585707
3- Could you provide more details about the preprocessing strategy that uses Grad-Cam to improve lung disease classification?
4- The article mentions the use of data augmentation strategies to address the unbalanced distribution of classes in the dataset. Can you explain how these strategies were implemented and their impact on the model's performance?
5- Dense blocks are mentioned as a component used to improve classification outcomes. Can you elaborate on what dense blocks are and how they contribute to the MixNet-LD system's effectiveness?
6- How does the MixNet-LD system compare to existing methods or models for lung disease diagnosis and severity categorization? Are there any notable advantages or limitations when compared to other approaches?
7- Can you elaborate on the practical implications of the MixNet-LD system for the field of medical image processing in clinical settings? How might this research impact real-world healthcare practices?
Further proofreading is required for the manuscript
Author Response

(The authors gave the same response as above.)

Round 2
Reviewer 1 Report
Check the correctness of data presented in Table 10.
Tables 9 and 11: use the same units of measurement for better comparison.
Algorithms 1 and 2: specify inputs and outputs.
Author Response
Dear and Respected Reviewer,
Thank you for your valuable comments. We have carefully addressed your valuable comments in the attached revised article draft.
With utmost respect,

Reviewer 2 Report
The authors have performed the revision excellently, and there are no further comments.
Author Response
Dear and Respected Reviewer,
Thank you again for your valuable comments and appreciation.
Best regards,
Reviewer 3 Report
Thanks to authors for their implementation
Author Response
Dear and Respected Reviewer,
Thank you.
Best regards,